# Attenuation of PITPNM1 Signaling Cascade Can Inhibit Breast Cancer Progression

**DOI:** 10.3390/biom11091265

**Published:** 2021-08-25

**Authors:** Zihao Liu, Yu Shi, Qun Lin, Wenqian Yang, Qing Luo, Yinghuan Cen, Juanmei Li, Xiaolin Fang, Wen G. Jiang, Chang Gong

**Affiliations:** 1Breast Tumor Center, Sun Yat-Sen Memorial Hospital, Sun Yat-Sen University, Guangzhou 510120, China; liuzh75@mail2.sysu.edu.cn (Z.L.); shiyu9712@126.com (Y.S.); linq27@mail2.sysu.edu.cn (Q.L.); ywq20541@126.com (W.Y.); luoq63@mail2.sysu.edu.cn (Q.L.); cenyinghuan@163.com (Y.C.); lijm95@mail.sysu.edu.cn (J.L.); fangxiaolin2021@126.com (X.F.); 2Guangdong Provincial Key Laboratory of Malignant Tumor Epigenetics and Gene Regulation, Sun Yat-Sen Memorial Hospital, Sun Yat-Sen University, Guangzhou 510120, China; 3Cardiff China Medical Research Collaborative, School of Medicine, Cardiff University, Heath Park, Cardiff CF14 4XN, UK; jiangw@cardiff.ac.uk

**Keywords:** breast cancer, PITPNM1, proliferation, metastasis

## Abstract

Phosphatidylinositol transfer protein membrane-associated 1 (PITPNM1) contains a highly conserved phosphatidylinositol transfer domain which is involved in phosphoinositide trafficking and signaling transduction under physiological conditions. However, the functional role of PITPNM1 in cancer progression remains unknown. Here, by integrating datasets of The Cancer Genome Atlas (TCGA) and Molecular Taxonomy of Breast Cancer (METABRIC), we found that the expression of PITPNM1 is much higher in breast cancer tissues than in normal breast tissues, and a high expression of PITPNM1 predicts a poor prognosis for breast cancer patients. Through gene set variation analysis (GSEA) and gene ontology (GO) analysis, we found PITPNM1 is mainly associated with carcinogenesis and cell-to-cell signaling ontology. Silencing of PITPNM1, in vitro, significantly abrogates proliferation and colony formation of breast cancer cells. Collectively, PITPNM1 is an important prognostic indicator and a potential therapeutic target for breast cancer.

## 1. Introduction

According to the 2020 cancer epidemiology statistics, breast cancer is the most common cancer worldwide among women with an estimated 2.3 million newly diagnosed cases [1,2]. Breast cancer becomes a major public health problem worldwide. Thus, uncovering new molecular mechanisms which are involved in the progression of breast cancer is of great clinical significance.

PITPNM1 (Phosphatidylinositol Transfer Protein Membrane Associated 1), also known as NIR2 (Pyk2 N-terminal domain-interacting receptor 2) and RDGB1 (Retinal Degeneration B Alpha 1), belongs to the subfamily of membrane-associated phosphatidylinositol transfer domain-containing proteins. PITPNM1 shares homology with Drosophila retina degeneration B (rdgB) protein [3,4,5]. PITPNM1, PITPNM2, and PITPNM3 were first found to be expressed in the retina and brain [6,7]. The genetic analysis of a small cohort of patients with autosomal dominant retinal dystrophy reveals that a point mutation c.1878G>C (p.Q626H) of PITPNM1′s homology, PITPNM3, will lead to retinal dystrophy disease [8]. These findings suggest that PITPNM1 might play a functional role in the phototransduction pathway and PITPNM1is considered as a candidate gene for human retina diseases. Recent studies suggest PITPNM1 can also be expressed in hair cells, adipose-derived stem cells, and hearing cells [9,10,11]. These studies, collectively, suggest that PITPNM1 plays a complex role in normal cells. Interestingly, cancer cells seem to hijack PITPNM1 to support their progression. It is reported that PITPNM1 is expressed in breast cancer cells and PITPNM1 promotes breast cancer migration, invasion, and epithelial-mesenchymal-transition (EMT) by activating the PI3K-AKT signaling pathway. These indicate that PITPNM1 potentially regulates the complex steps of cancer progression [12,13]. However, little attention has been drawn to PITPNM1 and its involvement in cancer progression. To reveal the interplay between PITPNM1 and the hallmark events of breast cancer, we performed data mining in The Cancer Genome Atlas (TCGA) and Molecular Taxonomy of Breast Cancer (METABRIC) to explore the association between the expression of PITPNM1 and clinicopathological data. Moreover, we carried out a series of in vitro tests by silencing PITPNM1 to determine whether PITPNM1 regulates breast cancer proliferation or not. Our results indicate that PITPNM1 plays an oncogenic role in breast cancer.

## 2. Materials and Methods

### 2.1. Cell Culture

Human breast cancer cell lines including SKBR3, MDA-MB-468, BT549, MDA-MB-231, T47D, ZR75-1, MCF-7, and BT474 were purchased from the American Type Culture Collection (ATCC, Manassas, VA, USA). All cells were cultured in Dulbecco’s Modified Eagle Medium (DMEM) (Gibco, Life Technologies, Grand Island, NY, USA) supplemented with 10% fetal bovine serum under 5% CO_2_ at 37 °C. All cells were confirmed by short tandem repeat DNA profiling and passaged for less than 6 months.

### 2.2. SiRNA Transfection

Breast cancer cells (5 × 10 per well) were seeded in a 6-well plate. Cells were cultured and transfected from 30–50% confluence. Two siRNAs targeting different sites of PITPNM1 (si-PITPNM1-1, sense: 5′-GCGGGCAAUACACACACAATT-3′, anti-sense: 5′-UUGUGUGUGUAUUGCCCGCTT-3′; si-PITPNM1-2, sense: 5′-GGAGAAAUUCUCCAUUGAATT-3′, anti-sense: 5′-UUCAAUGGAGAAUUUCUCCTT-3′, Genepharma, Shanghai, China) were, respectively, suspended in Opti-MEM and mixed with lipo3000. The mixture was added to plates and cultured for 48 h. After transfection, cells were harvested for further experiments.

### 2.3. CCK8 Cell Counting Assay

The CCK8 assay was used to assess the proliferation ability of cells. Cells (3 × 10^3^ per well for MDA-MB-231, 5 × 10^3^ per well for MCF-7 and SKBR3) were seeded in 96-well plates, incubated, and tested with CCK8. The CCK8 assay was tested daily for MDA-MB-231 cells and every other day for MCF-7 and SKBR3 cells. CCK8 reagent was added in a culture medium and incubated at 5% CO_2_ at 37 °C for 1 h. Absorbance was measured by the TCAN plate reader (TECAN Spark 10M, Tecan Group Ltd., Zürich, Switzerland) at 450 nm. Data represent the mean of three independent experiments.

### 2.4. Colony Formation Assay

Cells were harvested after transfection, 1 × 10^3^ to 2 × 10^3^ cells were seeded in 6-well plates and cultured at 5% CO_2_ at 37 °C for at least two weeks. After incubation, cells were fixed with 4% formaldehyde and stained with crystal violet. Data represent the mean of three independent experiments.

### 2.5. RNA Isolation and Real-Time Quantitative PCR

Total RNA was extracted using the RNA extraction kit (RC112-01, Vazyme, Nanjing, China) according to the manufacturer’s protocol. Total RNA was used to synthesize complementary DNA (cDNA) through RNA reverse transcriptional reagents (Vazyme, Nanjing, China). Real-time quantitative PCR (qPCR) was performed by using SYBR green reagent (Vazyme, Nanjing, China) according to the manufacturer’s protocol. The relative expression of genes was normalized to ACTB and calculated by the 2^−ΔΔCt^ method. Primers are listed as follows: ACTB sense: 5′-AGCGGGAAATCGTGCGTGAC-3′, ACTB anti-sense: 5′-CAGGAAGGAAGGCTGGAAGAGT-3′; PITPNM1 sense: 5′-GAGGAGTCTAGTGGTGAGGGC-3′, anti-sense: 5′-TTCGGGTGTAGGGGTAGGC-3′.

### 2.6. Data Extraction and Bioinformatics Analysis

The transcriptome data of the gene count matrix was downloaded from the TCGA data portal (https://portal.gdc.cancer.gov/, accessed on 8 March 2020). The METABRIC datasets were downloaded from cBioPortal (http://www.cbioportal.org/, accessed on 13 February 2021).The Broad Institute Cancer Cell Line Encyclopedia (CCLE) datasets were downloaded from the official website (https://portals.broadinstitute.org/ccle, accessed on 8 May 2020). Estrogen receptor (ER) and progesterone receptor (PR) status were determined by immunohistochemistry. Human epidermal growth factor receptor 2 (HER2) status was determined by immunohistochemistry and/or fluorescence in situ hybridization (FISH). The survival data of the TCGA breast cancer cohort were downloaded from the TCGA Pan-Cancer Clinical Data Resource. Survival analysis was performed by website tools (Kaplan-Meier plotter, http://kmplot.com/analysis/, accessed on 10 November 2020; bc-GenExMiner, http://bcgenex.ico.unicancer.fr/BC-GEM/, accessed on 23 April 2021). Differentially expressed genes were analyzed by DESeq2. The gene ontology of genes differentially expressed in the high PITPNM1 group and the low PITPNM1 group was analyzed by DAVID and gene set variation analysis (GSEA) [14,15]. Immune infiltration scores were analyzed by the CIBERSORTx website portal (https://cibersortx.stanford.edu/, accessed on 29 February 2021) [16].

### 2.7. Statistical Analysis

Statistical analysis was performed using IBM SPSS Statistical 25 and R Studio v5. The log2FC > 1.5 and *p* < 0.05 or log2FC < −1.5 and *p* < 0.05 were characterized as differentially expressed genes. A two-tail student *t*-test and two-tail one-way ANOVA with Tukey’s post hoc test were used to determine statistical significance between groups. K-M survival analysis was used to determine the prognosis of the high PITPNM1 and the low PITPNM1 groups with the higher quartile regarded as the cut-off value and their significance was calculated by the log-rank test.

## 3. Results

### 3.1. PITPNM1 Is Significantly Higher in Breast Cancer and Correlates with Poor Prognosis in Patients with Breast Cancer

We extracted two large-scale transcriptome datasets of breast cancer to determine the expression of PITPNM1. The mRNA of PITPNM1 of TCGA breast cancer samples was significantly higher in triple-negative cancer (TNBC, 1.64-fold, *p* < 0.0001, *n* = 160) as well as HER2 over-expression in breast cancer (1.99-fold, *p* < 0.0001, *n* = 33) and luminal A/B (1.55-fold, *p* < 0.0001, *n* = 711), compared to normal breast tissue (*n* = 113, Figure 1A). In breast cancer tissues of the TCGA datasets, the expression of PITPNM1 barely correlates with the breast cancer stage, ER status (*p* > 0.05), PR status (*p* > 0.05) or HER2 status (*p* > 0.05, Figure 1B,C). In METABRIC datasets, the expression of PITPNM1 in breast tissues is consistent with that seen in the TCGA datasets. The expression of PITPNM1 in TNBC (1.48-fold, *p* < 0.0001, *n* = 299), HER2 over-expression in breast cancer tissues (1.59-fold, *p* < 0.0001, *n* = 127), and luminal A/B (1.25-fold, *p* < 0.0001, *n* = 1464) is higher than that in normal breast tissues (*n* = 147, Figure 1D). The expression of PITPNM1 in stage 3 (*p* < 0.05) and stage 4 (*p* < 0.05) breast cancers is significantly higher than in stage 1 (Figure 1E). Compared with other subtypes, the level of PITPNM1 was higher in the ER-negative group, PR-negative group, and HER2-positive group (Figure 1F). Since PITPNM1 is much higher in breast cancer tissues than in normal breast tissues, we further explored the prognostic role of PITPNM1. By using the higher quartile, we separated breast cancer patients into two groups: the high expression group and the low expression group. We found high levels of PITPNM1 marginally associated with short disease-free survival (DFS) in the TCGA cohort (log-rank *p* = 0.1075), but not significantly correlated with overall survival (OS, Figure 1G). To further confirm whether high levels of PITPNM1 are associated with poor prognosis, three other independent cohorts of breast cancer patients were separated into two groups by the higher quartile, respectively. We found that higher expression of PITPNM1 was associated with poor OS in the METABRIC, SCAN-B, and Kaplan–Meier Plotter cohort (Figure 1H–J).

### 3.2. PITPNM1 Over-Expression Is Associated with Carcinogenesis Gene Ontology and Pathway

Since high expression of PITPNM1 predicts poor OS of patients with breast cancer as shown earlier, PITPNM1 could potentially promote breast cancer progression. In order to explore the potential molecular mechanisms underlying carcinogenesis of PITPNM1 in breast cancer, we employed GO term analysis, KEGG analysis, and GSEA analysis to determine the enrichment of cancer-related processes in the high PITPNM1 group and the low PITPNM1 group. Firstly, we analyzed the differentially expressed genes between the high PITPNM1 group and the low PITPNM1 group, defined by the higher quartile in the TCGA BRCA cohort. A total of 4543 differentially expressed genes were identified (Figure 2A, Appendix A).Through KEGG pathway analysis, we found that up-regulated genes in the high PITPNM1 group are mainly enriched in immune pathways such as: the T cell receptor signaling pathway, rheumatoid arthritis, primary immunodeficiency, and NK cell-mediated cytotoxicity, and cancer-related signaling pathways such as the NF-kappa B signaling pathway, the Jak-STAT signaling pathway, the chemokine signaling and cell adhesion molecules (Figure 2B, Appendix A). In contrast, down-regulated genes are mainly enriched in tight junction, taste transduction, steroid hormone biogenesis, salivary secretion, retinol metabolism, and so on (Figure 2C, Appendix A). Gene ontology analysis indicates that up-regulated genes are enriched in immune processes such as immune processes, inflammatory response, innate immune response, regulation of immune response, as well as T cell co-stimulation and cancer-related biological processes such as cell adhesion, positive regulation of cell proliferation, and cell differentiation (Figure 2D, Appendix A). As for the cellular counterpart, these genes are mainly enriched in the plasma membrane, the integral component of the membrane, and the extracellular region (Figure 2D, Appendix A). Molecular functions such as calcium ion binding, sequence-specific DNA binding, receptor binding, and so on, are mainly enriched by up-regulated genes. Down-regulated genes are not significantly associated with cancer progression since down-regulated genes are mainly enriched in multicellular organism development, the transport muscle contraction, and the chemical synaptic transmission for biological processes; the integral component of the membrane, the extracellular region, the extracellular space for the cellular counterpart, the G protein receptor activity, the olfactory receptor activity, actin-binding, and the structure molecule activity for molecular function (Figure 2E, Appendix A).

To strengthen the analysis of the DAVID database, we performed GSEA, which can reveal a great number of gene sets as well as their phenotypes enriched in the highPITPNM1group and the low PITPNM1 group. Interestingly, hallmarks of the T cell immune process such as the regulation of T cell activation and the activation of T cell proliferation, and hallmarks of cancer such as the regulation of cell-cell adhesion and the positive regulation of G1/S transition of the mitotic cell cycle, cell cycle check points are significantly enriched in the high PITPNM1 group, while negative regulation of the execution phase of apoptosis is enriched in the low PITPNM1 group (Appendix A).The top GSEA biological processes enriched in the high PITPNM1 group and the low PITPNM1 group are shown in Figure 2F.

Furthermore, we used the hallmarks of gene sets (h: hallmark gene sets, GSEA) which are gene signatures derived by many gene sets to represent well-defined biological states or processes to explore the potential biological processes enriched in the high PITPNM1 group as well as the low PITPNM1 group. Some cancer-related processes such as PI3K-AKT signaling, UV response signaling, the P53 pathway, and mitotic-spindle are enriched in the high PITPNM1 group while none of the gene sets are enriched in the low PITPNM1 group (Appendix A). The top one enrichment of GSEA (h: hallmark gene sets) analysis of the high PITPNM1 group is shown in Figure 2G.

### 3.3. Silencing of PITPNM1 Inhibits Proliferation of Breast Cancer Cells

PITPNM1 predicts the poor prognosis of breast cancer and is associated with carcinogenesis gene ontology. The role of PITPNM1 in breast cancer progression was further determined. Firstly, the expression of PITPNM1 in different breast cancer cells was explored in CCLE datasets (Figure 3A). In different molecular subtypes, the mean value of PITPNM1 expression was much higher in HER2 over-expression and TNBC than that in the luminal A/B subtype. To validate the expression of PITPNM1 in different breast cancer cell lines and a non-tumorigenic immortalized breast epithelial cell, the mRNA levels of PITPNM1 were determined in nine cell lines including HER2 over-expression, TNBC, luminal subtypes, and non-tumorigenic MCF-10A (Figure 3B). PITPNM1 is highly expressed in most breast cancer cell lines. To confirm these results of transcript analyses, the protein level of PITPNM1 was also tested. Consistent with the qPCR results, the protein level of PITPNM1 was higher in breast cancer cell lines (Figure 3C). From the cell lines tested, SKBR3, MDA-MB-231, MCF-7, and MCF-10A cells were chosen. Two siRNAs targeting PITPNM1 were used to create PITPNM1-silencing cell models. Transfection with PITPNM1 siRNAs significantly reduced its expression from 60–70% in four different cell lines (Figure 3D). The CCK8 assay indicated that the silencing of PITPNM1 significantly inhibited MDA-MB-231 proliferation (Figure 3E). The colony formation ability of MDA-MB-231 was decreased after PITPNM1 siRNAs transfection (Figure 3E). In addition, the luminal subtype cancer cell line MCF7, as well as the HER2 over-expression cell line SKBR3, showed the same response to treatment of PITPNM1 siRNAs transfection (Figure 3F,G). On the other hand, silencing PITPNM1 in non-tumorigenic MCF-10A had little impact on cell proliferation or cell viability (Figure 3H). This indicates that PITPNM1 is essential for breast cancer proliferation but is not required by MCF-10A. PITPNM1 is a potential oncogene. Experiments in vitro suggest that PITPNM1 promotes breast cancer proliferation and maintains breast cancer cell viability.

### 3.4. The Relationship between PITPNM1 Expression and T Cell Populations

Through GSEA analysis, we found the high PITPNM1 group can enrich T cell immune process ontology such as T cell activation and T cell differentiation. In order to explore the potential involvement of PITPNM1 in the T cell immune process, firstly, we calculated the absolute abundance of T cell sub-populations in breast cancer by CIBERSORTx, which is a confident analytical tool to estimate the abundance of member cell types in cancer micro-environments based on bulk RNA sequencing. Activated CD4 positive memory T cells, CD8 positive follicular helper T cells, and regulatory T cells are significantly dysregulated in breast cancer tissues (Figure 4A). CD8 positive T cells, resting CD4 positive memory T cell, CD4 positive naive T cells, and gamma delta T cells showed no statistical difference between normal breast tissue and breast cancer tissue (Figure 4A). Since PITPNM1 is significantly up-regulated in breast cancer tissue (Figure 4B), we analyzed the correlation coefficient between PITPNM1 and the abundance of T cell sub-populations, which is significantly dysregulated in cancer tissues. PITPNM1 is correlated with regulatory T cells (r = 0.28, *p* < 0.0001, Figure 4C, Appendix A). These findings suggest PITPNM1 is potentially involved in breast cancer progression by regulating regulatory T cell infiltration and regulatory T cell function.

## 4. Discussion

In the present study, we have revealed that PITPNM1 is associated with breast cancer prognosis. Through bioinformatics analysis, we found that PITPNM1 is associated with the carcinogenesis pathway and T cell immune process. The silencing of PITPNM1 inhibits the proliferation and colony formation ability of MDA-MB-231, MCF-7, and SKBR3. Moreover, PITPNM1 positively correlates with regulatory T cell infiltration. These results indicate that PITPNM1 could promote breast cancer progression by regulating breast cancer proliferation and the T cell immune process.

Previous studies have shown that PITPNM1 is an important regulator in maintaining the normal physiological functions of cells. PITPNM1 is expressed in retinal cells, adipose-derived stem cells, hearing cells, as well as hair cells, and is required for the normal function of these cells [7,9,10]. The dysregulation of PITPNM1 impairs signaling transduction and phosphatidylinositol transference, which in turn leads to abnormal functions of certain cells. It is reported that PITPNM1 is dysregulated in tissues from hearing-loss mouse models and Otx2 conditional knockout mice [11,17]. Moreover, PITPNM1 is regarded as a phosphoinositide trafficking and signaling transduction regulator [18,19]. It not only transfers phosphoinositide but also delivers phosphatidic acid between the endoplasmic reticulum and plasma membrane [18,20]. These findings highlight the important roles of PITPNM1 in homeostasis. However, few studies reveal the oncogenic roles of PITPNM1.

Of note, by integrating two large-scale breast cancer RNA sequence datasets, we found PITPNM1 was over-expressed in breast cancer tissue. We also found ahigh level of PITPNM1 can predict poor the prognosis of breast cancer patients. These results reveal the potential involvement of PITPNM1 in breast cancer progression. Interestingly, a recent study has provided some evidence for this link and showedPITPNM1 can promote breast metastasis by regulating breast cancer migration, invasion, and epithelial–mesenchymal-transition. The silencing of PITPNM1 will result in the impairment of migration as well as invasion [12]. In addition to regulating cell functions pertinent to metastasis, we found that PITPNM1 could regulate cell proliferation as well.

PITPNM1 was also shown to regulate the infiltration of regulatory T cells. Regulatory T cells play a tumor-promotional role in the micro-environment since regulatory T cells can migrate into cancer and suppress the antitumor effects of lymphocytes such as CD8 positive cytotoxic T cells [21,22,23]. Interestingly, through GSEA analysis, we found that the high PITPNM1 group could enrich T cell immune process ontology such as T cell activation and T cell differentiation. It raises the possibility that PITPNM1 potentially promotes breast cancer progression by regulating T cell immune processes. By using a highly confident immune abundance estimation tool, CIBERSORTx, we analyzed T cell subset abundance in bulk RNA sequence datasets. In addition, we found PITPNM1 was significantly correlated with regulatory T cells infiltration but is barely correlated with the infiltration of naive CD4 positive T cells or gamma delta T cells. Consistent with PITPNM1′s cancer-promotional roles, PITPNM1 promotes breast cancer progression by regulating regulatory T cell infiltration and inhibiting antitumor T cell immune processes.

## 5. Conclusions

PITPNM1, a phosphoinositide trafficking and signal transduction regulator, can promote breast cancer progression by regulating breast cancer proliferation and regulatory T cell infiltration. Therefore, PITPNM1 is proven to be an actionable target for breast cancer treatment.

## Figures and Tables

**Figure 1 biomolecules-11-01265-f001:**
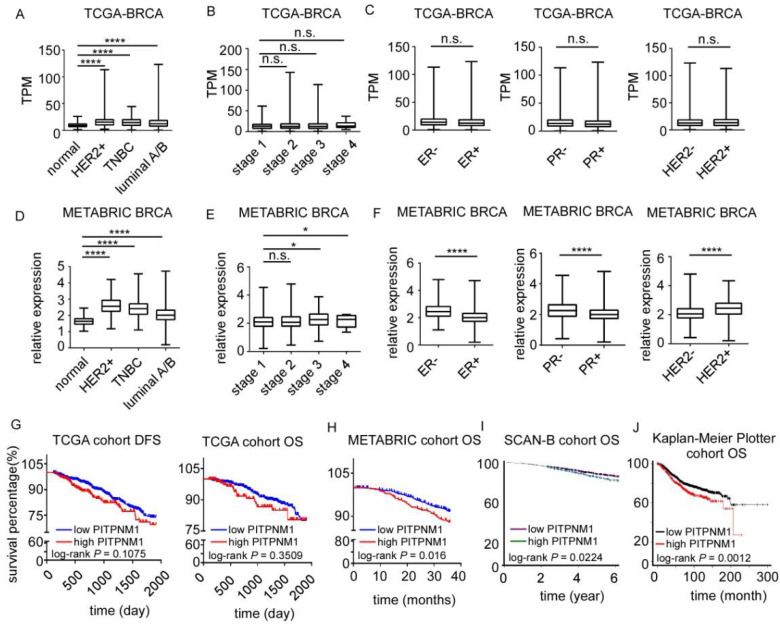
Prognostic value of PITPNM1 in breast cancer. (**A**) The expression of PITPNM1 in different molecular subtypes of breast cancer of TCGA cohort. Triple-negative cancer, TNBC, *n* = 160; human epidermal growth factor receptor 2 over-expression, HER2+, *n* = 33; luminal A/B, *n* = 711; normal breast tissue *n* = 113. **** *p* < 0.0001. (**B**) The expression of PITPNM1 in different stages of breast cancer of TCGA cohort. (**C**) The association of PITPNM1 with estrogen receptor (ER) status, progesterone receptor (PR) status, and HER2 status in TCGA cohort. n.s., not significant, *p* > 0.05. (**D**) The expression of PITPNM1 in different molecular subtypes of breast cancer of METABRIC cohort. TNBC, *n* = 299; HER2+, *n* = 127; luminal A/B, *n* = 1464; normal breast tissue *n* = 147. **** *p* < 0.0001. (**E**) The expression of PITPNM1 in different stages of breast cancer of METABRIC cohort. n.s., not significant, *p* > 0.05; * *p* < 0.05. (**F**) The association of PITPNM1 with estrogen receptor (ER) status, progesterone receptor (PR) status, and HER2 status in METABRIC cohort. **** *p* < 0.0001. (**G**) The prognosis of PITPNM1 in TCGA breast cancer cohort. Disease-free survival time, DFS; overall survival, OS. (**H**) The prognosis of PITPNM1 in METABRIC cohort. (**I**) The prognosis of PITPNM1 in SCAN-B cohort. (**J**) The prognosis of PITPNM1 in Kaplan–Meier cohort.

**Figure 2 biomolecules-11-01265-f002:**
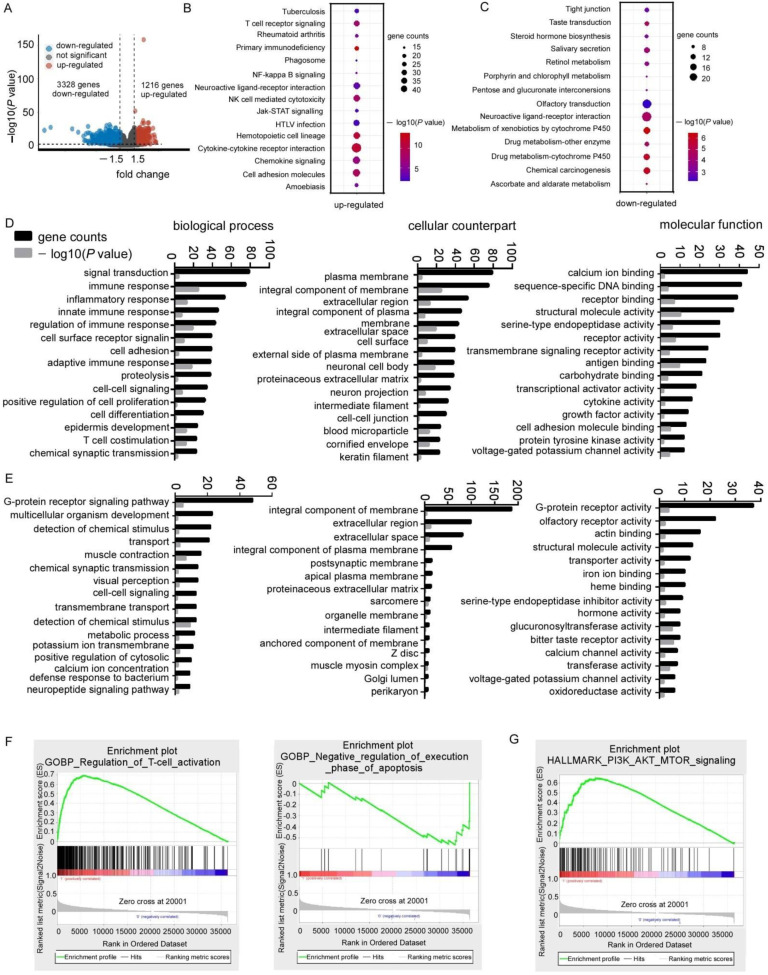
Gene ontology, KEGG pathway, and gene set enrichment analysis between PITPNM1 high expression group and PITPNM1 low group. (**A**) Volcano plot of differentially expressed genes between PITPNM1 high expression group and PITPNM1 low expression group in TCGA cohort. (**B**) KEGG pathway of up-regulated genes between the high PITPNM1 group and the low PITPNM1 group. (**C**) KEGG pathway of down-regulated genes between the high PITPNM1 group and the low PITPNM1 group. (**D**) Biological process, cellular component, and molecular function ontology analysis of up-regulated genes. (**E**) Biological process, cellular component, and molecular function ontology analysis of down-regulated genes. (**F**) GSEA (C5 gene ontology sets) analysis of signatures of PITPNM1 high expression group and PITPNM1 low expression group. (**G**) GSEA (h: hallmark gene sets) analysis of signatures of PITPNM1 high expression group and PITPNM1 low expression group.

**Figure 3 biomolecules-11-01265-f003:**
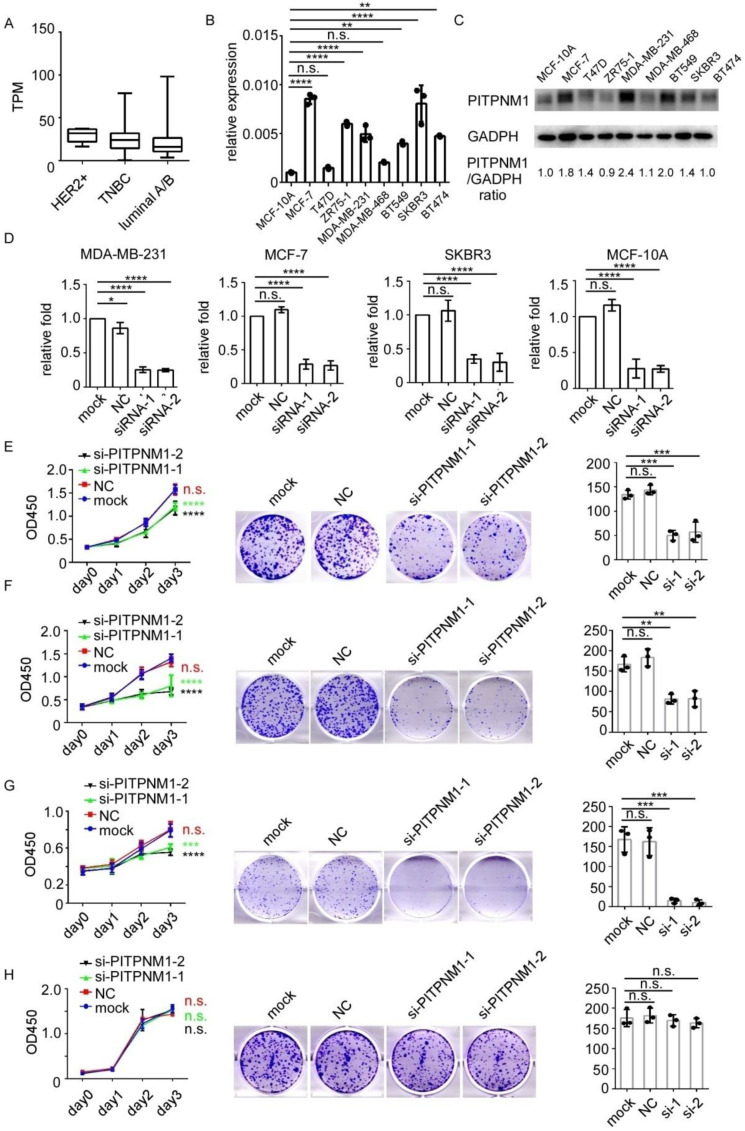
Silencing of PITPNM1 inhibits breast cancer proliferation. (**A**) The expression of PITPNM1 in different breast cancer subtypes. (**B**) The expression of PITPNM1 in breast cancer cell line validated by qPCR. n.s., not significant, *p* > 0.05; * *p* < 0.05; ** *p* < 0.01; *** *p* < 0.001; **** *p* < 0.0001. (**C**) The expression of PITPNM1 in breast cancer cell line validated by Western Blot. (**D**) siRNAs transfection efficiency validated by qPCR in four cell lines. n.s., not significant, *p* > 0.05, * *p* < 0.05; **** *p* < 0.0001. (**E**) The proliferation ability of MDA-MB-231 determined by the CCK8 assay and colony formation. n.s., not significant, *p* > 0.05; * *p* < 0.05; *** *p* < 0.001; **** *p* < 0.0001. (**F**) The proliferation ability of MCF-7 determined by the CCK8 assay and colony formation. n.s., not significant, *p* > 0.05; * *p* < 0.05; ** *p* < 0.01; **** *p* < 0.0001. (**G**) The proliferation ability of SKBR3 determined by the CCK8 assay and colony formation. n.s., not significant, *p* > 0.05; * *p* < 0.05; *** *p* < 0.001; **** *p* < 0.0001. (**H**) The proliferation ability of MCF-10A determined by the CCK8 assay and colony formation. n.s., not significant, *p* > 0.05.

**Figure 4 biomolecules-11-01265-f004:**
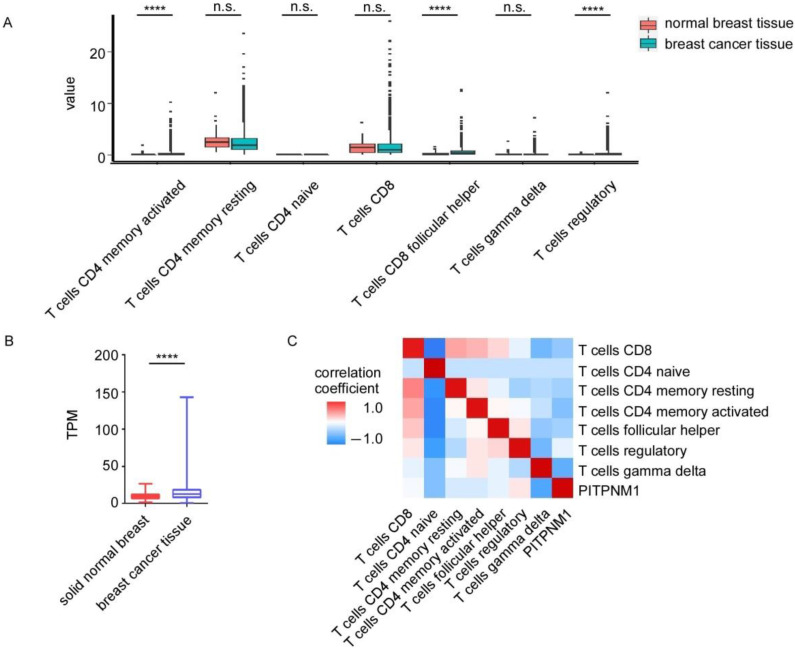
The relationship between PITPNM1 and T cell sub-population abundance. (**A**) The abundance of T cell subtypes in TCGA cohort estimated by CIBERSORTx. *t*-test; n.s., not significant, *p* > 0.05; **** *p* < 0.0001. (**B**) The expression of PITPNM1 in TCGA breast cancer cohort. Normal breast, *n* = 113; breast cancer tissue, *n* = 1104. **** *p* < 0.0001. (**C**) Heatmap of Spearman correlation coefficient between PITPNM1 and T cell subtypes.

## Data Availability

The data presented in this study are available upon request from the corresponding authors.

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
