# Peer review of "Attenuation of PITPNM1 Signaling Cascade Can Inhibit Breast Cancer Progression"

_biomolecules, 2021, doi:10.3390/biom11091265_

Round 1
Reviewer 1 Report
The manuscript by Liu et al analyzed the role of Phosphatidylinositol transfer protein membrane associated 1 (PITPNM1) in breast cancer. PITPNM1 mRNA expression was examined in different patient databases and correlated with the prognosis value in patients. Differentially expressed genes associated with PITPNM1 expression in the database were identified and further analyzed by several pathway enrichment. The authors also validated the biological role of PITPNM1 by siRNA depletion in multiple breast cancer cell lines. In addition, the correlation between PITPNM1 and T cell population in breast cancer was examined. Overall, these data suggested that PITPNM1 is potentially associated with breast cancer progression. However, this study heavily relies on the analyses based on public datasets while the functional study is limited, which hampers the impact of the study. Below are the concerns from the reviewer for this study.
Major concern:
- All the current analyses were basically based on the mRNA expression, to further demonstrate PITPNM1 gene product is highly expressed in breast cancer comparing with normal samples. PITPNM1 protein levels should be examined either in breast cancer and normal epithelial mammary cells or normal/tumor tissue samples from patients with breast cancer.
- How to explain the different results regarding PITPNM1 mRNA expression across subtypes (Figure 1A and D), stage (1B and E), and ER/PR/HER2 status (1C and F)?
- Normal-like or true normal samples should be added in the METABRIC data as showed in Figure 1D.
- The overall biological value of data in Figure 2 seems to be limited. First, not sure how authors defined the PITPNM1 expression with high and low categories, and very few differentially expressed genes were identified between the high and low groups; second, the pathway analysis, GO or GESA analysis does not distinguish these up- or down-regulated genes showed in the volcano plot, therefore, it is hard to determine how these enriched pathways changed between the two groups; third, there are no significant pathways related to breast cancer carcinogenesis enriched in either of the three analyses, so the current data can not sufficiently support PITPNM1 over-expression is associated with carcinogenesis.
- The value of data from Figure 3A is very limited, what does the data suggest?
- It is good to show phenotypic validation data with three highly expressed cell lines, this is clear to show PITPNM1 is important for breast cancer proliferation. To rule out the possibility of PITPNM1 is an essential gene, authors should choose breast cancer cell lines with low expression or normal epithelial breast cell lines to perform PITPNM1 knockdown and proliferation assay.
Minor concerns:
- Figure 1D was not referred to in the text and there was no description for this data. Please add.
- What's the cut-off of the volcano plot? The ticks are not clearly labeled in the x- and y-axis in Figure 2B. In addition, the number of up- and down-regulated genes should be shown in the plot.
- Please explain why all GSEA pathways have the same NES score and p-value as shown in Figure 2E.
- Please add statistical analysis for Figures 3B, G, H, and I.
Author Response
Reviewer 1
We are very grateful to the reviewer for the constructive comments. These comments have been fully addressed in the revised version. The following are the details of the modifications.
The manuscript by Liu et al analyzed the role of Phosphatidylinositol transfer protein membrane associated 1 (PITPNM1) in breast cancer. PITPNM1 mRNA expression was examined in different patient databases and correlated with the prognosis value in patients. Differentially expressed genes associated with PITPNM1 expression in the database were identified and further analyzed by several pathway enrichment. The authors also validated the biological role of PITPNM1 by siRNA depletion in multiple breast cancer cell lines. In addition, the correlation between PITPNM1 and T cell population in breast cancer was examined. Overall, these data suggested that PITPNM1 is potentially associated with breast cancer progression. However, this study heavily relies on the analyses based on public datasets while the functional study is limited, which hampers the impact of the study. Below are the concerns from the reviewer for this study.
Major concern:
- All the current analyses were basically based on the mRNA expression, to further demonstrate PITPNM1 gene product is highly expressed in breast cancer comparing with normal samples. PITPNM1 protein levels should be examined either in breast cancer and normal epithelial mammary cells or normal/tumor tissue samples from patients with breast cancer.
Thanks for the reviewer’s constructive suggestions. We further tested the mRNA level of PITPNM1 in a non-tumorigenic epithelial mammary cell line MCF-10A and find that the mRNA level of PITPNM1 is much higher in breast cancer cell lines when compared to non-tumorigenic MCF-10A. Besides, we examine the protein level of PITPNM1 in different breast cancer cell lines as well as non-tumorigenicepithelial mammary cell line MCF-10A as suggested. The protein level is consistent to the mRNA level in most cell lines. We revised our manuscript as follow “To validate the expression of PITPNM1 in different breast cancer cell lines and a non-tumorigenic immortalized breast epithelial MCF-10A, nine cell lines including HER2 over-expression, TNBC, luminal subtypes and non-tumorigenic were tested (figure 3B). The expression of PITPNM1 in most breast cancer cell lines is much higher than non-tumorigenic MCF-10A. To confirm these results, the protein level of PITPNM1 is also tested. Consist to the qPCR results, the protein level of PITPNM1 is much higher in breast cancer cell lines (figure 3C).” (Page 6)
- How to explain the different results regarding PITPNM1 mRNA expression across subtypes (Figure 1A and D), stage (1B and E), and ER/PR/HER2 status (1C and F)?
This is indeed a very interesting comment. With the TCGA BRCA and METABRIC BRCA datasets, we noticed the same issue with regard to PITPNM1 mRNA expression across tumor stage, ER, PR, HER2 status. Whilst TCGA is a next-generation sequencing based project, METABRIC is a microarray based project. The different detection method will result in the different detection of relative expression level of PITPNM1. Besides, TCGA BRCA includes different races across different ages and these patients from TCGA received different therapy, while METABRIC collects patients passed selection criteria from tumor banks in the UK and Canada. Different cohort with different clinical background as well as different detection methods would contribute to some of these variations. This partly contributes to the different results regarding PITPNM1 mRNA level across subtypes, stages, ER/PR/HER2 status. Since PITPNM1 plays a functional roles in breast cancer proliferation and silence of PITPNM1 have little impact on non-tumorigenic mammary epithelial, we think PITPNM1 is a potential target for breast cancer treatment. This is a challenging area for the database and for scientific investigations. We will further carry on our study on the PITPNM1 in our cohort and integrate more RNA-seq data as well as microarray data to minish variations caused by different studies.
- Normal-like or true normal samples should be added in the METABRIC data as showed in Figure 1D.
Thanks for the reviewer’s suggestion, we have added the expression of PIPTNM1 of normal breast tissue of METABRIC dataset in Figure 1D and mentioned the results in manuscript “In METABRIC datasets, the expression of PITPNM1 in breast tissues is consistent to TCGA datasets. The expression of PITPNM1 in TNBC (1.48 fold, P < 0.0001, N=299), HER2 over-expression breast cancer (1.59 fold, P < 0.0001, N = 127) and luminal A/B (1.25 fold, P < 0.0001, N = 1464) is higher than normal breast tissue (N = 147, figure 1D).” (Page 3)
- The overall biological value of data in Figure 2 seems to be limited. First, not sure how authors defined the PITPNM1 expression with high and low categories, and very few differentially expressed genes were identified between the high and low groups; second, the pathway analysis, GO or GESA analysis does not distinguish these up- or down-regulated genes showed in the volcano plot, therefore, it is hard to determine how these enriched pathways changed between the two groups; third, there are no significant pathways related to breast cancer carcinogenesis enriched in either of the three analyses, so the current data can not sufficiently support PITPNM1 over-expression is associated with carcinogenesis.
Thanks for the reviewer’s comments and suggestions. We defined the high expression and low expression by higher quartile which is determined by K-M survival curves analysis. PITPNM1 high and PITPNM1 low group defined by higher quartile show statistically significant of OS in TCGA, metabric, SCAN-B and Kaplan-Meier Plotter cohort (Figure 1G-1J). High expression of PITPNM1, which is determined by higher quartile, will predict poor OS of breast cancer patients, thus we compared differentially expressed genes between high PITPNM1 group and low PITPNM1 group which is define by higher quartile to explore up-regulated as well as down-regulated genes which may associated with poor prognosis of breast cancer. We added the definition of high and low PITPNM1 group in the “2.7 Statistical analysis” Method part “K-M survival curves analysis were used to determine prognosis of high PITPNM1 and low PITPNM1 groups with higher quartile regarded as cut-off value and their significance were calculated by log-rank test.” (Page 3). “Since high expression of PITPNM1 will predict poor OS of breast cancer patients, we analyzed the differentially expressed genes between high PITPNM1 and low PITPNM1 group which is defined by higher quartile in TCGA BRCA cohort to explore the potential molecular mechanism underlying carcinogenesis of PITPNM1 in breast cancer.” (Page 4). We thanks for the reviewer’s suggestion on differentially expressed genes. In the old version of our manuscript, we used log2(fold change) > 1.5 or log2(fold change) < -1.5 and P< 0.05 to define differentially expressed genes between high and low PITPNM1 groups. Under the review’s reminder, it is more appropriate to define differentially expressed genes by fold change > 1.5 and P < 0.05 or fold change < -1.5 and P < 0.05, which is consistence to experimental practice and is also supported by other studies[1-3]. After reset the criteria, a total of 4543 differentially expressed genes was identified. We revised our manuscript as follow “A total of 4543 differentially expressed genes was identified (figure 2A, supplementary table 1).” (Page 4).
The suggestions about GO and GESA analysis was also fully appreciated. We re-analyzed differentially expressed genes and re-analyzed GO term by dividing them into up-regulated group and down-regulated group. With the reviewer’s suggestion, we found that up-regulated genes of high expression of PITPNM1 group mainly are enriched in immune process and some cancer-related GO term such as: cell adhesion and positive regulation of cell proliferation which is consistent with GSEA analysis. Besides, we found that KEGG pathways analysis of up-regulated genes mainly are enriched in immune pathways as well as cancer progression related pathways. We revised our manuscript as follow “Through KEGG pathway analysis, we found that up-regulated genes in high PITPNM1 group mainly enrich in immune processes such as: T cell receptor signaling pathway, rheumatoid arthritis, primary immunodeficiency and NK cell mediated cytotoxicity and cancer related signaling pathways such as: NF-kappa B signaling path-way, Jak-STAT signaling pathway, Chemokine signaling and cell adhesion molecules (figure 2B, supplementary table 2). On the other hand, down-regulated genes mainly enriched in tight junction, taste transduction, steroid hormone biogenesis, salivary secretion, retinol metabolism and so on (figure 2C, supplementary table 3). Gene ontology analysis indicates that up-regulated genes are enriched in immune processes such as: immune processes, inflammatory response, innate immune response, regulation of immune response as well as T cell costimulation and cancer related biological processes such as: cell adhesion, positive regulation of cell proliferation and cell differentiation (figure 2D, supplementary table 2). As for cellular counterpart, these genes mainly enriches in plasma membrane, integral component of membrane, extracellular region and so on (figure 2D, supplementary table 2). Molecular function such as: calcium ion binding, sequence specific DNA binding, receptor binding and so on were mainly enriched by up-regulated genes. Down-regulated genes not significantly associated with cancer progression, since down-regulated genes mainly enriches in multicellular organism development, transport muscle contraction and chemical synaptic transmission for biological process; integral component of membrane, extracellular region, extracellular space for cellular counterpart and G protein receptor activity, olfactory receptor activity, actin binding, structure molecule activity for molecular function (figure 2E, supplementary table 3).” (Page 5-Page 6). Since GSEA analysis is based on the relative expression of genes (Transcripts Per Million) andthe input data required normalized gene expression matrix of all genes (GSEA document: https://www.gsea-msigdb.org/gsea/doc/GSEAUserGuideFrame.html), we did not divide differentially expressed genes into up-regulated group or down-regulated group but we added gene sets enriched in PITPNM1 low group in our revised manuscript (gene sets enrich in high PITPNM1 group NES > 0, gene sets enrich in low PITPNM1 group NES < 0).As suggested, we revise our manuscript as follow “Interestingly, hallmarks of T cell immune process such as: regulation of T cell activation, activation of T cell proliferation and hallmarks of cancer such as: regulation of cell-cell adhesion, positive regulation of G1/S transition of mitotic cell cycle, cell cycle checkpoint is significantly enriched in PITPNM1 high expression group, while negative regulation of execution phase of apoptosis is enriched in PITPNM1 low group (supplementary table 4). Top one GSEA biological processes enriched in high PITPNM1 group and low PITPNM1 group are showed in figure 2F. Besides, we use hallmarks of gene sets (h: hallmark gene sets, GSEA) which are gene signatures derived by many gene sets to represent well-defined biological states or processes to explore the potential biological processes enriched in high PITPNM1 group as well as low PITPNM1 group. Some cancer related processes such as: PI3K-AKT signaling, UV response signaling, P53 pathway and mitotic-spindle are enriched in high PITPNM1 group while none of gene sets enriched in low PITPNM1 group (supplementary table 5). Top one enrichment of GSEA (h: hallmark gene sets) analysis of high PITPNM1 is showed in figure 2G.” (Page 6).
With the reviewer’s suggestion, our manuscript is much improved. We find the results of GO analysis is more consistent to the GSEA analysis in our new manuscript. After revision, we found that immune processes, cancer related ontology such as: positive regulation of cell proliferation as well as cell adhesion and cancer related pathways such as: NF-kappa B signaling path-way, Jak-STAT signaling pathway, chemokine signaling and cell adhesion molecules can be enriched in PITPNM1 high group. These results indicate that PITPNM1 potentially plays a complicated roles in physiological and pathological processes. To further validate the analysis, we found silencing of PITPNM1 will impact on cell proliferation. These confirmed over-expression of PITPNM1 is associated with carcinogenesis.
- The value of data from Figure 3A is very limited, what does the data suggest?
Thanks for the reviewer’s concerns about the meaning of figure 3A. Figure 3A showed the levels of PITPNM1 is dynamic in different breast cancer cell lines from CCLE datasets in the old version of manuscript. In addition, figure 3A and figure 3B share the same results. Considering the mislead of figure 3A and the repetition of figure 3A and figure 3B, we deleted the misleading graph in the new version of manuscript.
- It is good to show phenotypic validation data with three highly expressed cell lines, this is clear to show PITPNM1 is important for breast cancer proliferation. To rule out the possibility of PITPNM1 is an essential gene, authors should choose breast cancer cell lines with low expression or normal epithelial breast cell lines to perform PITPNM1 knockdown and proliferation assay.
Thanks for the reviewer’s suggestions. As suggested, we silence PITPNM1 in non-tumorigenic MCF-10A and perform proliferation assay of siRNA transfected MCF-10A cells. We found knockdown of PITPNM1 have little impact on MCF-10A proliferation (figure 3H). It is a interesting result that PITPNM1 is essential for breast cancer proliferation, but is not required by MCF-10A. This indicates that PITPNM1 is a potential oncogene. As suggested, we revised our manuscript as follow “On the other hand, knockdown of PITPNM1 in non-tumorigenic MCF-10A has little impact on cell proliferation as well as cell viability (figure 3H). This indicates that PITPNM1 is essential for breast cancer proliferation but is not required by MCF-10A. PITPNM1 is a potential oncogene. ” (Page 6)
Minor concerns:
- Figure 1D was not referred to in the text and there was no description for this data. Please add.
Thanks for the reviewer’s suggestions. As suggested, we revised our manuscript as follow “In METABRIC datasets, the expression of PITPNM1 in breast tissues is consistent to TCGA datasets. The expression of PITPNM1 in TNBC (1.48 fold, P < 0.0001, N=299), HER2 over-expression breast cancer (1.59 fold, P < 0.0001, N = 127) and luminal A/B (1.25 fold, P < 0.0001, N = 1464) is higher than normal breast tissue (N = 147, figure 1D).” (Page 3)
- What's the cut-off of the volcano plot? The ticks are not clearly labeled in the x- and y-axis in Figure 2B. In addition, the number of up- and down-regulated genes should be shown in the plot.
Thanks for the reviewer’s suggestions. The cut-off of the volcano plot is fold change > 1.5 and P < 0.05 or fold change < -1.5 and P < 0.05. A total of 4543 differentially expressed genes were identified. Among them, 1218 gene is up-regulated and 3328 genes is down-regulated (figure 2A, supplementary table 1). We revised our manuscript and have indicated these values in the new version of our manuscript.
- Please explain why all GSEA pathways have the same NES score and p-value as shown in Figure 2E.
Thanks for the reviewer’s concerns about GSEA analysis. The top 20 enrichment of GSEA analysis is statistically close, for instance, NES of “GOBP_REGULATION_OF_T_CELL_ACTIVATION”is 2.39768, while NES of “GOBP_ACTIVATED_T_CELL_PROLIFERATION” is 2.3919873 (supplementary table 4). The NES score as well as p-value is provided in the supplementary table 4 and supplementary table 5.
- Please add statistical analysis for Figures 3B, G, H, and I.
Thanks for the reviewer’s suggestions. We have revised our manuscript accordingly.
Ref.
[1]. Warden C D, Yuan Y C, Wu X. Optimal calculation of RNA-Seq fold-change values[J]. International Journal of Computational Bioinformatics and In Silico Modeling, 2013, 2(6): 285-292.
[2]. Zhang S, Cao J. A close examination of double filtering with fold change and t test in microarray analysis[J]. BMC bioinformatics, 2009, 10(1): 1-9.
[3]. McCarthy D J, Smyth G K. Testing significance relative to a fold-change threshold is a TREAT[J]. Bioinformatics, 2009, 25(6): 765-771.
Reviewer 2 Report
While this article has potential, there are some major flaws in its presentation (e.g., some figures are too difficult to see and decipher) and with English (grammar and spelling errors). I highly recommend that you submit this paper to a native English speaker before presenting it again for publication. There are also concerns about some conclusions made based on the data presented. Details are listed below.
Grammar
line 34 change 'mechanisms which involve in' to mechanisms which are involved in.
line 45 change 'These suggest the complicate' to These suggest a complex
line 50 change 'regulates the complicate steps of' to regulates the complex steps of
line 54 change clinical pathologic to one word clinicopathological, insert the word 'and' after association and before significance and the word 'the' between of and PITPNM1 gene.
line 83 change Each experiment was repeat independently for three times to ....Data represent the mean of 3 independent experiments.
line 140 insert we explored between tissue and the prognostic role and delete 'is also being explored' at the end of the sentence.
line 181 change 'strength the analysis' to strengthen the analysis
line 213 insert the word cells after MCF-7 and add an 's' to level
the word dysregulated is mispelled throughout the text (lines 246, 272, and 274)
line 273 change 'which is turn' to which in turn
line 278 remove the letter 's' from the word highlight
line 289 change PITPNM1 also show its potential to PITPNM1 also showed is potential
In the methods section, be sure to make key numbers either superscripts or subscripts (sections 2.3 and 2.4)
Science
1. Remember use of siRNA technology is silencing and not knockdown so change the wording accordingly.
2. At the end of section 2.5 there is a sentence that indicates that Primers are listed as follows but nothing is there! Please list them.
3. Line 143 you mention that you found high level of PITPNM1 marginally significantly predicted short disease free survival time in TCGA cohort, but not significantly correlates with overall survival referring to figure 1G but it appears that neither reaches statistical significance.
4. Figure 2A is a bit confusing. What is it truly trying to show? Figure 2C is very difficult to see and the GSEA in 2E is virtually impossible to see. These must be made larger.
5. Gene ontology analysis shows which genes are enriched but what does this mean? It is important to explain how this supports the hypothesis that PITPNM1 promotes breast cancer progression and thus must be targeted.
Author Response
Reviewer 2
We are very grateful for the comments. We have fully addressed the comments in our new version of the manuscript. The following are the list of modifications.
While this article has potential, there are some major flaws in its presentation (e.g., some figures are too difficult to see and decipher) and with English (grammar and spelling errors). I highly recommend that you submit this paper to a native English speaker before presenting it again for publication. There are also concerns about some conclusions made based on the data presented. Details are listed below.
Grammar
line 34 change 'mechanisms which involve in' to mechanisms which are involved in.
line 45 change 'These suggest the complicate' to These suggest a complex
line 50 change 'regulates the complicate steps of' to regulates the complex steps of
line 54 change clinical pathologic to one word clinicopathological, insert the word 'and' after association and before significance and the word 'the' between of and PITPNM1 gene.
line 83 change Each experiment was repeat independently for three times to ....Data represent the mean of 3 independent experiments.
line 140 insert we explored between tissue and the prognostic role and delete 'is also being explored' at the end of the sentence.
line 181 change 'strength the analysis' to strengthen the analysis
line 213 insert the word cells after MCF-7 and add an 's' to level
the word dysregulated is mispelled throughout the text (lines 246, 272, and 274)
line 273 change 'which is turn' to which in turn
line 278 remove the letter 's' from the word highlight
line 289 change PITPNM1 also show its potential to PITPNM1 also showed is potential
In the methods section, be sure to make key numbers either superscripts or subscripts (sections 2.3 and 2.4)
Thanks for the reviewer’s suggestions. We have corrected the grammar errors as suggested.
Science
- Remember use of siRNA technology is silencing and not knockdown so change the wording accordingly.
We are grateful for the suggestions. As suggested, we have changed the wording from “knockdown” to “silencing”.
- At the end of section 2.5 there is a sentence that indicates that Primers are listed as follows but nothing is there! Please list them.
Thanks for the reviewer’s suggestions. We have added the primers in our manuscript as follow “Primers are listed as follow: ACTB sense: 5’-AGCGGGAAATCGTGCGTGAC-3’, ACTB anti-sense: 5’- CAGGAAGGAAGGCTGGAAGAGT-3’; PITPNM1 sense: 5’- GAG-GAGTCTAGTGGTGAGGGC-3’, anti-sense: 5’- TTCGGGTGTAGGGGTAGGC-3’”(Page 3)
- Line 143 you mention that you found high level of PITPNM1 marginally significantly predicted short disease free survival time in TCGA cohort, but not significantly correlates with overall survival referring to figure 1G but it appears that neither reaches statistical significance.
Thanks for the reviewer’s suggestions. We mentioned that high level of PITPNM1 marginally significantly predicted short disease free survival time with the log-rank P = 0.1075. It is obviously not statistical significance and that is why we use “marginally significantly” rather “statistically significantly”. With the reviewer’s suggestions, we think “marginally significantly” might mislead the reader, so we change our manuscript as follow “We found high level of PITPNM1 marginally associates short disease free survival time (DFS) in TCGA cohort (log-rank P = 0.1075), but not significantly correlates with overall survival (OS, figure 1G)”. Besides, in order to validate the “marginal” results, we used another three datasets and found that high level of PITPNM1 statistically significantly predict poor prognosis of breast cancer (figure 1H-1J).
- Figure 2A is a bit confusing. What is it truly trying to show? Figure 2C is very difficult to see and the GSEA in 2E is virtually impossible to see. These must be made larger.
Thanks for the reviewer’s suggestions. Figure 2A is the heatmap of differently expressed genes between high PITPNM1 group and low PITPNM1 group in TCGA cohort. In the new version, we have deleted this figure. As suggested, we make GSEA analysis results larger in figure 2E.
- Gene ontology analysis shows which genes are enriched but what does this mean? It is important to explain how this supports the hypothesis that PITPNM1 promotes breast cancer progression and thus must be targeted.
Thanks for the reviewer’s suggestions. To make it clear why we use gene ontology, KEGG pathway as well as GSEA analysis, we revised our manuscript as follow “Since high expression of PITPNM1 will predict poor OS of breast cancer patients, PITPNM1 potentially promotes breast cancer progression. In order to explore the potential molecular mechanisms underlying carcinogenesis of PITPNM1 in breast cancer, we employ GO term analysis, KEGG analysis and GSEA analysis to determine the enrichment of cancer related processes in high PITPNM1 group and low PITPNM1 group. Firstly, we analyzed the differentially expressed genes between high PITPNM1 and low PITPNM1 group which is defined by higher quartile in TCGA BRCA cohort. A total of 4543 differentially expressed genes was identified (figure 2A, supplementary table 1). By integrate up-regulated gene sets in the high PITPNM1 group, we can find out the abnormally activated gene ontology as well as pathway related to high expression of PITPNM1.” (Page 4-Page 5)
Round 2
Reviewer 1 Report
This revised manuscript has been largely improved according to reviewers' comments. The authors have addressed all my concerns in detail. However, further language editing is needed before publication.
Author Response
Reviewer 1
We are very grateful to the reviewer for the constructive comments. These comments have been fully addressed in the revised version.
This revised manuscript has been largely improved according to reviewers' comments. The authors have addressed all my concerns in detail. However, further language editing is needed before publication.
Thanks for the reviewer’s suggestion. We have double checked our manuscript, corrected grammar errors and edited our manuscript. Many thanks!
Reviewer 2 Report
It seems that you have taken care to correct a number of the suggestions I made for this paper. It seems that the science is solid and your data is consistent with the conclusions drawn in your manuscript. There are still some moderate changes needed in the grammar/syntax before the paper should be accepted for publication. I have noted some of these changes below.
- There are a number of times when the word 'the' was missing throughout the paper and should be inserted. This occurs a number of times when you discuss and compare high vs low PITPNMI groups. Rather than stating high PITPNM1 and low PITPNM1 groups you should state the high PITPNM1 and the low PITPNM1 groups. Otherwise, insert 'the' in the following places listed below:
- in Line 105 CCK8 assay were tested ...should read The CCK8 assay was tested.
- In Line 107 measure by TCAN should read measure by the TCAN...
- Line 138 using RNA extraction kit should be read as using the RNA extraction kit
- In lines 362-363 you should insert the word 'the' before the different pathways listed so it reads the NF-kappa B signaling pathway, the Jak-STAT signaling pathway and so forth
- Lines 371-372 change in plasma membrane to in the plasma membran, the integral component of the membrane and the extracellular region.
- In line 736 change associated with carcinogenesis pathway to associated with the carcinogenesis pathway
- Line 779 change role in microenvironment.... to role in the microenvironment.
Other grammatical suggestions
- In the first paragraph the sentence should read According to the 2020 cancer epidemiology statistics, breast cancer is the most common cancer worldwide among women with an estimated 2.3 newly diagnosed cases. The words in bold should be substituted for what is currently written.
- The last part of the final sentence in the first paragraph should read..developing new forms of targeted therapy rather than developing new targets for therapy.
- In line 77 change we carried out series of in vitro tests to we carried out a series of in vitro...
- Change the sentence in lines 134-145 from Each experiments was repeat independently for three times to Data represent the mean of three independent experiments.
- Change lines 207-208 to read short disease free survival [delete the word time] in the cohort(log-rank P = 0.1075), but does not significantly correlate [delete the letter s] with overall survival.....
- In line 209 change sentence to read high levels [add s to level] of PITPNM1 are [rather than is] associated with poor prognosis....
- In line 357 change By integrating (rather than integrate) up-regulated gene sets........
- Line 376 mainly enrich [change from enriches] in multicellular organism
- Line 386 change cell cycle checkpoint are [change from is] significantly enriched in....
- Line 396 Add the word The to start the sentence so it reads The top one enrichment of GSEA analysis of high PITPNM1 is shown [not showed] in figure 2G.
- Line 501 change Consist to the qPCR results..... to Consistent with the qPCR results.
- Line 532-533 change These findings suggest PITPNM1 potentially involves in.... to These findings suggest PITPNM1 is potentially involved in..
- Line 748 change hearing-loss mice models to hearing-loss mouse models
- Line 749 add the article/word 'a' to state regarded as a phosphoinositide trafficking and signal [delete the 'ing'] transduction regulator
- Line 790 delete the 's' from the word cells in the phrase regulatory T cells.
- Line 795 change the last sentence to read Therefore, PITPNM1 may prove to be an actionable target for breast cancer treatment.
Author Response
Reviewer 2
We are very grateful to the reviewer for the constructive comments. These comments have been fully addressed in the revised version.
It seems that you have taken care to correct a number of the suggestions I made for this paper. It seems that the science is solid and your data is consistent with the conclusions drawn in your manuscript. There are still some moderate changes needed in the grammar/syntax before the paper should be accepted for publication. I have noted some of these changes below.
- There are a number of times when the word 'the' was missing throughout the paper and should be inserted. This occurs a number of times when you discuss and compare high vs low PITPNMI groups. Rather than stating high PITPNM1 and low PITPNM1 groups you should state the high PITPNM1 and the low PITPNM1 groups. Otherwise, insert 'the' in the following places listed below:
- in Line 105 CCK8 assay were tested ...should read The CCK8 assay was tested.
- In Line 107 measure by TCAN should read measure by the TCAN...
- Line 138 using RNA extraction kit should be read as using the RNA extraction kit
- In lines 362-363 you should insert the word 'the' before the different pathways listed so it reads the NF-kappa B signaling pathway, the Jak-STAT signaling pathway and so forth
- Lines 371-372 change in plasma membrane to in the plasma membran, the integral component of the membrane and the extracellular region.
- In line 736 change associated with carcinogenesis pathway to associated with the carcinogenesis pathway
- Line 779 change role in microenvironment.... to role in the microenvironment.
Other grammatical suggestions
- In the first paragraph the sentence should read According to the 2020 cancer epidemiology statistics, breast cancer is the most common cancer worldwideamong women with an estimated 2.3 newly diagnosed cases. The words in bold should be substituted for what is currently written.
- The last part of the final sentence in the first paragraph should read..developing new forms of targeted therapy rather than developing new targets for therapy.
- In line 77 change we carried out series of in vitro tests to we carried out a series of in vitro...
- Change the sentence in lines 134-145 from Each experiments was repeat independently for three times to Data represent the mean of three independent experiments.
Change lines 207-208 to read short disease free survival [delete the word time] in the cohort(log-rank P = 0.1075), but does not significantly correlate [delete the letter s] with overall survival.....
In line 209 change sentence to read high levels [add s to level] of PITPNM1 are [rather than is] associated with poor prognosis....
In line 357 change By integrating (rather than integrate) up-regulated gene sets........
Line 376 mainly enrich [change from enriches] in multicellular organism
Line 386 change cell cycle checkpoint are [change from is] significantly enriched in....
Line 396 Add the word The to start the sentence so it reads The top one enrichment of GSEA analysis of high PITPNM1 is shown [not showed] in figure 2G.
Line 501 change Consist to the qPCR results..... to Consistent with the qPCR results.
Line 532-533 change These findings suggest PITPNM1 potentially involves in.... to These findings suggest PITPNM1 is potentially involved in..
Line 748 change hearing-loss mice models to hearing-loss mouse models
Line 749 add the article/word 'a' to state regarded as a phosphoinositide trafficking and signal [delete the 'ing'] transduction regulator
Line 790 delete the 's' from the word cells in the phrase regulatory T cells.
Line 795 change the last sentence to read Therefore, PITPNM1 may prove to be an actionable target for breast cancer treatment.
Thanks for the reviewer’s suggestion. The modifications mentioned by the reviewer were addressed. Besides, we have double checked our manuscript, corrected grammar errors and edited our manuscript. Please find the revised manuscript. Many thanks!